# Generalization properties of neural network approximations to frustrated magnet ground states

Tom Westerhout[1 ✉], Nikita Astrakhantsev[2,3,4 ✉], Konstantin S. Tikhonov [5,6,7 ✉], Mikhail I. Katsnelson[1,8] & Andrey A. Bagrov[1,8,9 ✉]

Neural quantum states (NQS) attract a lot of attention due to their potential to serve as a very expressive variational ansatz for quantum many-body systems. Here we study the main factors governing the applicability of NQS to frustrated magnets by training neural networks to approximate ground states of several moderately-sized Hamiltonians using the corresponding wave function structure on a small subset of the Hilbert space basis as training dataset. We notice that generalization quality, i.e. the ability to learn from a limited number of samples and correctly approximate the target state on the rest of the space, drops abruptly when frustration is increased. We also show that learning the sign structure is considerably more difficult than learning amplitudes. Finally, we conclude that the main issue to be addressed at this stage, in order to use the method of NQS for simulating realistic models, is that of generalization rather than expressibility.

[1] Institute for Molecules and Materials, Radboud University, Heyendaalseweg 135, 6525 AJ Nijmegen, The Netherlands. [2] Physik-Institut, Universität Zürich, Winterthurerstrasse 190, CH-8057 Zürich, Switzerland. [3] Moscow Institute of Physics and Technology, Institutsky lane 9, 141700 Dolgoprudny, Russia. [4] Institute for Theoretical and Experimental Physics NRC Kurchatov Institute, 117218 Moscow, Russia. [5] Skolkovo Institute of Science and Technology, 143026 Skolkovo, Russia. [6] Institut für Nanotechnologie, Karlsruhe Institute of Technology, 76021 Karlsruhe, Germany. [7] Landau Institute for Theoretical Physics RAS, 119334 Moscow, Russia. [8] Theoretical Physics and Applied Mathematics Department, Ural Federal University, 620002 Yekaterinburg, Russia. [9] Department of Physics and Astronomy, Uppsala University, Box 516, SE-75120 Uppsala, Sweden. ✉email: t.westerhout@student.science.ru.nl; nikita.astrakhantsev@phystech.edu; tikhonov@itp.ac.ru; andrey.bagrov@physics.uu.se

Following fascinating success in image and speech recognition tasks, machine-learning (ML) methods have recently been shown to be useful in physical sciences. For example, ML has been used to classify phases of matter[1], to enhance quantum state tomography[2,3], to bypass expensive dynamic ab initio calculations[4], and more[5]. Currently, artificial neural networks (NNs) are being explored as variational approximations for many-body quantum systems in the context of variational Monte Carlo (vMC) approach. vMC is a well-established class of methods suitable for studying low-energy physics of many-body quantum systems with a more than 50-year history[6]. A vast variety of trial wave functions have been suggested in different contexts. One of the simplest choices is mean-field form of the wave function which can be enriched by explicit account for particle–particle correlation[7–9] and generalized to include many variational parameters[10–13]. Certain tensor network variational ansätze, e.g. matrix product states[14], do not require stochastic Monte Carlo sampling and are thus amenable to exact optimization. The common shortcoming of all these methods is that the trial functions are tailored to a concrete model of interest and often require some prior knowledge about structure of the ground state (such as short-range entanglement for MPS methods) or intuition which helps constrain the optimization landscape (e.g. approximate understanding of nodal surfaces for Quantum Monte Carlo methods[15]). However, in many cases our prior intuition can be insufficient or unreliable. This poses a natural question whether a more generic ansatz that can efficiently approximate ground states of many-body systems could exist.

A novel and fresh look at this problem was given in ref. [16], where the traditional vMC optimization approach was hybridized with ML. A simple yet very unrestricted variational ansatz that inherits the structure of a certain neural network—restricted Boltzmann machine (RBM)—was suggested. For the test cases of one-dimensional and two-dimensional Heisenberg and transverse field Ising models, it was demonstrated that, optimizing this ansatz with the stochastic reconfiguration (SR) scheme[17], one could achieve high accuracy in approximating ground states of systems of up to hundreds of spins, sometimes outperforming the state-of-the-art methods.

In subsequent years, a number of new variational neural quantum states (NQS) have been suggested and their properties were thoroughly analyzed. Among other important discoveries, it was realized that even the simplest RBMs with polynomial number of parameters have rich enough structure to host volume law entanglement[18,19], indicating that NQS are more flexible than, for instance, tensor networks[20]. Recently, RBM representation for open quantum systems has been formulated[21–23]. Hybrid wave functions, combining properties of RBMs and more traditional pair product wave functions, were demonstrated to significantly reduce relative energy error of variational ground state of two-dimensional Fermi–Hubbard model[24] and to enhance the accuracy of Gutzwiller-projected wave functions in frustrated magnets[25]. An algorithm for computing the spectrum of low-lying excited states has been suggested[26], opening a route to studying finite-temperature phenomena with NQS (see also ref. [27]). However, it also became evident that NQS must not be perceived as a magic bullet in the area of strongly correlated quantum systems[28]. Although a variational wave function with a network structure may be able to approximate the ground state really well, in some cases the desired point in the space of variational parameters can be hard to reach, and learning algorithm hits a saddle point before approaching the solution. This results in a large relative energy error and a low overlap between the NQS and the actual exact ground state, making the obtained solution almost useless for computing physical observables. This problem is particularly pronounced for systems where the energy gap

between the ground state and the first excited state is very small, like for frustrated spin systems such as $J_1 - J_2$ antiferromagnetic Heisenberg model on square lattice[29], or the Fermi–Hubbard model away from the neutrality point[30].

While it can be proven mathematically that NNs can in principle approximate any smooth function to arbitrary accuracy[31], it might require an impractically large number of parameters. Thus an important feature of any ansatz is its expressibility—a potential capacity to represent a many-body wave function with high accuracy using a moderate number of parameters[32–34], and so far, significant effort has been put into the search for NQS architectures that possess this property[3,35]. At the same time, there is another issue that is not widely discussed in this context—the generalization properties of an ansatz. To illustrate this aspect, it is instructive to consider the problem of fitting a known wave function by a certain ansatz. For a sufficiently large quantum system, even evaluation of the cost function (which depends on the ansatz parameters and measures the fit quality) and its gradients may become impossible as it requires summing over a very large number of terms in the Hilbert space. One may hope that, by instead evaluating the cost function on a smaller set of the Hilbert space basis, sampled in a certain way, one will eventually approach point of optimality close to the actual solution of the full optimization problem. This is not guaranteed at all, and is exactly what is known as generalization property in the ML context. This issue is also very important in the variational optimization scheme. In vMC, an ansatz is adjusted iteratively in a certain way, so that it is expected that the system ends up in the lowest energy state allowed by the form of the ansatz[17,36,37]. At each step of this iterative procedure, one has to evaluate the change of the trial wave function parameters. This relies on MC sampling from basis of the Hilbert space of the model, and for large systems the total number of samples is negligibly small in comparison with the dimension of the Hilbert space. Hence, it is of crucial importance for the ansatz to accurately generalize onto a larger subspace that was not sampled in the course of learning and correctly estimate phases and amplitudes of the wave function on the full set of basis vectors.

Although the generalization issue concerns both phases and amplitudes of the wave function coefficients, it turns out that these two components behave differently in this respect. Already from the first works in the field, it seemed plausible that effectiveness of NN as variational ansatz is somehow connected to the sign structure of the models. For instance, in ref. [16], even for the unfrustrated Heisenberg antiferromagnet on a square lattice, the Hamiltonian must first be brought into stoquastic (sign-definite) form by a unitary transformation in order to reduce noise and attain proper level of convergence (see also ref. [38]). As another example, let us note that in recent study[37] it was stressed that biasing the NQS ansatz with certain predefined (heuristic) sign structures is very important for performance of the method.

In this paper, we study generalization properties of NNs[39,40] in the context of approximating the eigenstates of large quantum systems paying special attention to the sign structure. By running numerical experiments, we shall demonstrate that it is indeed the lack of sign structure generalization that prevents a neural quantum state from learning the wave function, even though expressibility of the corresponding ansatz could be good enough. To do that, we focus on the antiferromagnetic Heisenberg model on square, triangular, and Kagome lattices with competing interactions. First, we solve each of the models using exact diagonalization. Then, with the exact ground state as a target, we use supervised learning to train the NQS. During the training procedure, NNs are shown only a tiny fraction of the ground state (which is chosen by sampling from the probability distribution $\propto |\psi_i|^2$). Quality of the approximation is then assessed on the

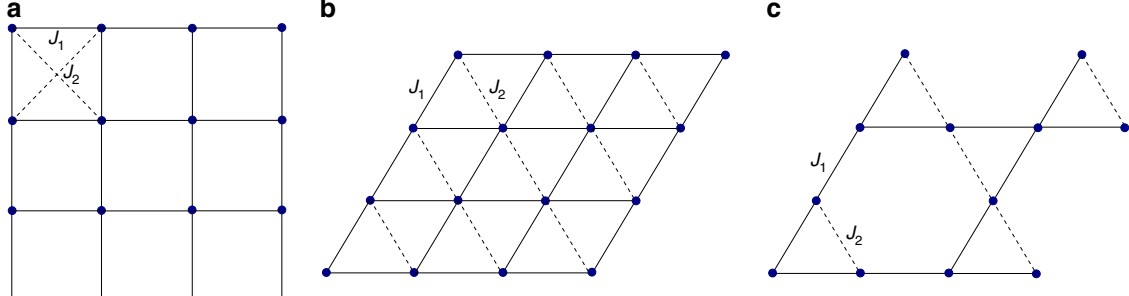

**Fig. 1 Lattices considered in this work.** We studied three frustrated antiferromagnetic Heisenberg models: **a** next-nearest neighbor $J_1-J_2$ model on square lattice; **b** anisotropic nearest-neighbor model on triangular lattice; **c** spatially anisotropic Kagome lattice. In all cases $J_2 = 0$ corresponds to the absence of frustration.

remaining part of the Hilbert space basis which we call test dataset. We tune both the training dataset size as well as the degree of frustration (controlled by $J_2/J_1$), and show that when the models are interpolated between unfrustrated and fully frustrated regimes, networks' generalization abilities change in a non-trivial way, with sign structure becoming very difficult to learn in certain cases. This motivates a search for NQS architectures which generalize better.

## Results

**Setup and main findings**. We consider several antiferromagnetic spin models described by the Heisenberg Hamiltonian:

$$\hat{H} = J_1 \sum_{\langle a,b \rangle} \hat{\boldsymbol{\sigma}}_a \otimes \hat{\boldsymbol{\sigma}}_b + J_2 \sum_{\langle\langle a,b \rangle\rangle} \hat{\boldsymbol{\sigma}}_a \otimes \hat{\boldsymbol{\sigma}}_b \ , \tag{1}$$

where for each lattice geometry, the first sum is taken over the unfrustrated sublattice (solid lines in Fig. 1), and the second sum is taken over the sublattice that brings in frustrations (dashed lines in Fig. 1). Namely, we consider $J_1-J_2$ model on a square lattice[41–43] and the nearest-neighbor antiferromagnets on spatially anisotropic triangular[44] and Kagome[45,46] lattices. These models are known to host spin liquid phases in certain domains of $J_2/J_1$, to which we further refer as frustrated regions.

For every model, its ground state belongs to the sector of minimal magnetization, thus the dimension of the corresponding Hilbert space is $K = C_{[N/2]}^N$ (where $N$ is the number of spins). It is convenient to work in the basis of eigenstates of $\hat{\sigma}^z$ operator: $|\mathcal{S}\rangle \sim |\uparrow\downarrow \ldots \downarrow\uparrow\rangle$. In this basis the Hamiltonian $\hat{H}$ is real-valued. The ground state is thus also real-valued, and every coefficient in its basis expansion is characterized by a sign $s_i = \text{sign}(\psi_i)$ (instead of a continuous phase):

$$|\Psi_{\text{GS}}\rangle = \sum_{i=1}^K \psi_i |\mathcal{S}_i\rangle = \sum_{i=1}^K s_i |\psi_i| |\mathcal{S}_i\rangle. \tag{2}$$

We have analyzed how NNs learn ground state structures of these models. In what follows, we will be mainly speaking about periodic clusters of 24 spins, since all the effects are clear already in that case, but will also provide results for 30-spin clusters, and some data for a 6-by-6 periodic square lattice. Effective dimension of a 24-spin system Hilbert space in the zero-magnetization sector is $d = C_{12}^{24} \simeq 2.7 \cdot 10^6$. Our main results seem universal for all studied models and architectures and can be summarized in the following four statements:

(i) Generalization from a relatively small subset of Hilbert space basis of the wave function sign structure is not granted even when the ansatz is able to express the ground

state with high accuracy. Very well known to ML practitioners, this fact is also valid for spin systems, in both frustrated and ordered regimes.

(ii) Construction and training of a network to achieve good generalization, a task which is relatively simple in the ordered phase, becomes much harder upon approaching the frustrated regime.

(iii) Quality of generalization depends on the size of training set in an abrupt way exhibiting a sharp increase at some $\varepsilon_{\text{train}} = \frac{\text{Training dataset size}}{\text{Hilbert space dimension}}$.

(iv) Generalization of wave function amplitudes turns out to be a substantially easier task than generalization of signs.

In the remaining part of this section we explain the findings in more detail using Kagome lattice as an example. We focus on a two-layer dense neural network architecture. For results for other models and detailed comparison of different architectures we refer the reader to Supplementary Notes 1 and 2.

**Generalization of sign structure**. Upper row on Fig. 2 illustrates both points i and ii. Here, we use a small subset (1%) of the Hilbert space basis to train the NN and then evaluate how well it predicts the sign structure on the remaining basis vectors unavailable to it during training. To assess the quality of generalization we use overlap between the exact ground state and the trial state. The latter is defined as a state with amplitudes taken from exact diagonalization and sign structure encoded in a NN (the sign is chosen by following the most probable outcome according to the NN). Consider, for example, panel **c**, where generalization quality for Kagome model is shown as a function of $J_2/J_1$. It is known[46] that Kagome model hosts a frustrated regime for $0.51 \lesssim J_2/J_1 \lesssim 1.82$. Strikingly, this phase transition shows itself as a sharp decrease of overlap around the value $J_2/J_1 \approx 0.51$. As one may expect, the frustrated regime is characterized by very intricate sign distribution leading to a drastic reduction in the overlap. For the square and the triangular lattices (Fig. 2a, b), generalization quality behaves somewhat differently. For $J_1-J_2$ model on the square lattice, instead of a sharp transition, it exhibits a large but smooth dip in frustrated regime ($0.4 < J_2/J_1 < 0.6$). On the triangular lattice, the minimum is reached slightly before approaching the transition point ($J_2/J_1 \approx 1.25$). However, for all three models we see that behavior of generalization quality reflects very well the known phase transitions with generalization being easy in ordered phases and becoming notoriously hard in disordered phases. Note also that different NNs may generalize very differently: in particular, as shown for the square and triangular lattices of Fig. 2, dips in performance of convolutional NNs are much smaller than for dense networks. Such good performance is most likely due to the fact that our

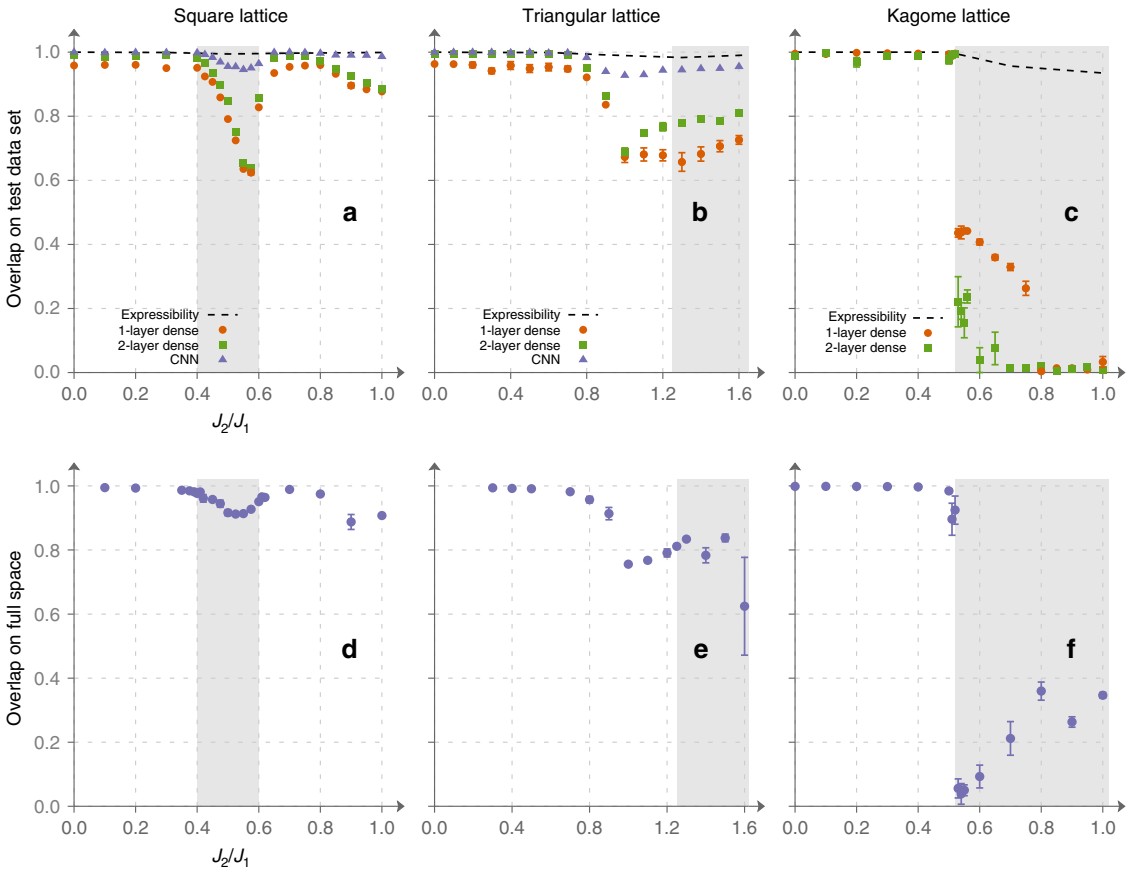

**Fig. 2 Optimization results for 24-site clusters obtained with supervised learning and stochastic reconfiguration.** Subfigures **a**–**c** were obtained using supervised learning of the sign structure. Overlap of the variational wave function with the exact ground state is shown as function of $J_2/J_1$ for square **a**, triangular **b**, and Kagome **c** lattices. Overlap was computed on the test dataset (not included into training and validation datasets). Note that generalization is poor in the frustrated regions (which are shaded on the plots). 1-layer dense, 2-layer dense, and convolutional neural network (CNN) architectures are described in Supplementary Note 1. Subfigures **d**–**f** show overlap between the variational wave function optimized using Stochastic Reconfiguration and the exact ground state for square, triangular, and Kagome lattices, respectively. Variational wave function was represented by two two-layer dense networks. A correlation between generalization quality and accuracy of the SR method is evident. On this figure, as well as on all the subsequent ones (both in the main text and Supplementary Notes 1 and 2), error bars represent standard error (SE) obtained by repeating simulations multiple times.

implementation of CNNs accounts for translational symmetry (see Supplementary Note 1 for an in-depth explanation of used NN architectures).

We believe that experiments of this kind would help to choose proper architectures to be used in vMC methods such as SR. In SR scheme, parameter updates are calculated using a small (compared to the Hilbert space dimension) set of vectors sampled from the probability distribution proportional to $|\psi|^2$. This closely resembles the way we choose our training dataset. Moreover, SR does not optimize energy directly, rather at each iteration it tries to maximize the overlap between the NQS $|\Psi\rangle$ and the result of its imaginary time evolution $(1 - \delta t \hat{H})|\Psi\rangle$. Hence, even though our supervised learning scheme and SR differ drastically, their efficiencies are strongly related. To make this correlation more apparent, we have performed several vMC experiments for 24-spin clusters. In the second row of Fig. 2, we provide results of SR simulations for different values of $J_2/J_1$. One can see that the two learning schemes follow very similar patterns.

Let us now turn to observation (iii). As we have already mentioned, it is very important to distinguish the ability to represent the data from generalization. In the context of NQS, the former means that a NN is able to express complex quantum states well if training was conducted in a perfect way. For clusters of 24 spins we have trained the networks on the

entire ground state and found that expressibility of the ansätze is not an issue—we could achieve overlaps above 0.94 for all values of $J_2/J_1$ (dashed lines in the upper row of Fig. 2). We believe that this result holds true for larger clusters though we could not verify this: for 30 spins (Hilbert space dimension $\sim 1.5 \times 10^8$) training the network on the entire set of basis vectors is too resource demanding. However, high expressibility does not automatically make a neural network useful if it cannot generalize well. To make the boundary more clear, we study how generalization quality changes when size of the training dataset is increased. Results for Kagome lattice are shown in Fig. 3. Interestingly, even in the frustrated regime ($J_2 = 0.6$) it is possible to generalize reasonably well from a relatively small subset of the basis states, but the required $\varepsilon_{train}$ becomes substantially larger than in the magnetically ordered phase. Most importantly, the ability of the NN to generalize establishes in an abrupt manner contrary to more typical smooth behavior observed in statistical models of learning[47–49]. Another interesting feature is saturation of the overlap at large $\varepsilon_{train}$ which can be observed in larger systems (see Supplementary Figs. 5 and 7). In the system of 24 spins, it is hard to see this plateau as it requires too large $\varepsilon_{train}$, such that all relevant basis vectors end up in the training dataset, and overlap computed on the rest of the basis is meaningless.

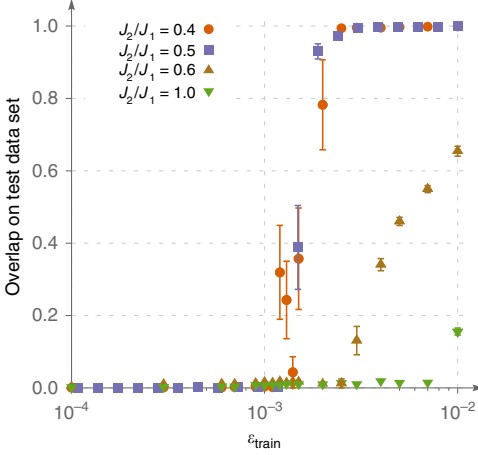

**Fig. 3 Dependence of generalization quality on the size of the training dataset.** Results are shown for 24-site Kagome model for $J_2/J_1 =$ 0.4, 0.5, 0.6, 1.0. $\varepsilon_{train}$ denotes the fraction of the Hilbert space basis used for training. Generalization quality is measured by overlap between the variational and the exact states computed on the test dataset.

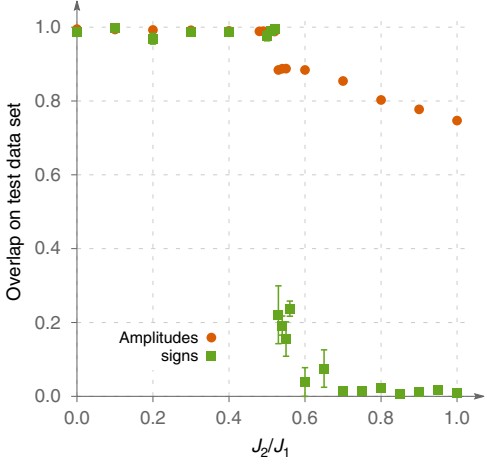

**Fig. 4 Generalization of signs and amplitudes.** We compare generalization quality as measured by overlap for learning the sign structure (red circles) and amplitude structure (green squares) for 24-site Kagome lattice for two-layer dense architecture. Note that both curves decrease in the frustrated region, but the sign structure is much harder to learn.

**Generalization of amplitudes.** In our discussion up to this point, we concentrated entirely on the quality of generalization of the wave function sign structure. One may wonder whether it is indeed the signs rather than amplitudes, which are responsible for the difficulty of learning the wave function as a whole (this possibility has been discussed in the context of state tomography in ref. [2]). To prove this statement, we conduct the following analysis. In the context of learning, overlap between a trial wave function and the target state can be used to characterize the effectiveness of NNs in two different ways. First, one can fix the amplitudes of the wave function and use a NN to learn the signs. This produces a trial wave function $\psi_{sign}$. Alternatively, one can fix the sign structure, and encode the amplitudes in a NN to get a trial wave function $\psi_{amp}$. Clearly, the accuracy of $\psi_{amp}$ and $\psi_{sign}$ will depend on the relative complexity of learning amplitudes and signs of the wave function coefficients. We illustrate statement (iv) with Fig. 4, where we use overlap to compare the quality of generalization of signs and amplitudes (using, again, 1% of the basis for training). Upon increase of $J_2$, one moves from a simple

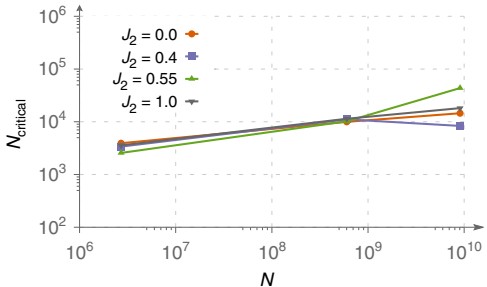

**Fig. 5 Scaling of the critical dataset size for the square lattice.** Training dataset size which is required for non-zero generalization is shown as a function of the Hilbert space dimension. One can see that it scales relatively slowly.

ordered phase to frustrated regime where overlap drops sharply. Although the generalization of both signs and amplitudes becomes harder at the point of phase transition $J_2/J_1 = 0.51$, drop in the sign curve is much larger, and at even higher $J_2$ the quality of the learned states becomes too poor to approximate the target wave function. At the same time, even deeply in the frustrated regime generalization of amplitudes, given the exact sign structure, leads to a decent result. Moreover, generalization quality of amplitudes does not drop abruptly when $\varepsilon_{train}$ is decreased, remaining non-zero on very small datasets (see Supplementary Fig. 3). These observations suggest that it is indeed the sign part of the wave function that becomes problematic for generalization in frustrated region. One should keep in mind that difficult to learn sign structure is not directly related to the famous Quantum Monte Carlo sign problem. For example, Fig. 2 shows that in $J_2 \to 0$ limit of $J_1-J_2$ model, networks have no trouble learning the sign structure even though in $\hat{\sigma}^z$ basis, there is sign problem since we are not applying Marshall's Sign Rule.

**Larger clusters.** So far we have been exemplifying our results on 24-spin clusters, and it is interesting to see whether the main observations hold for larger Hilbert spaces as well. Most of the computations that we performed can be repeated for lattices of 30 spins. Even bigger systems become too resource demanding and require a more involved algorithm implementation. Nevertheless, for the square lattice of 36 spins (6-by-6), we managed to compute dependency of generalization on the training dataset size for several values of $J_2/J_1$. For the detailed analysis of 30-spin clusters we refer the reader to Supplementary Note 2. One can see that all the conclusions remain valid—behavior of the generalization quality as function of $J_2/J_1$ is very similar to that for 24-spin clusters, and the dependence on $\varepsilon_{train}$ exhibits a sharp transition.

What is especially interesting is that the critical size of the training dataset required for non-zero generalization seems to scale relatively slowly with the system size. In Fig. 5, for the case of the square lattice, we show the critical size of the training dataset as a function of the Hilbert space dimension $K$. It turns out, that when one goes from 24 spins ($K \simeq 2.7 \times 10^6$) to 36 spins ($K \simeq 9 \times 10^9$), it is sufficient to increase the training dataset just by a factor of 10. This gives us hope that reasonable generalization quality can be achieved for even larger systems.

## Discussion

In this paper, we have analyzed the ability of NNs to generalize many-body quantum states from a small number of basis vectors to the whole Hilbert space. The main observation we made is that for all models we have considered, quality of generalization of the

ground state sign structure falls off near quantum phase transitions and remains low in the frustrated regimes.

We have demonstrated that generalization may indeed be an essential factor that is likely responsible for spoiling the convergence of NQS in a number of physically interesting cases, such as frustrated quantum spin systems. Our main conclusion which is qualitatively valid for all the studied models and NN architectures is that a NN struggles to generalize the distribution of signs in the ground state of a many-body system with competing interactions in the regime of strong frustrations if the training is done on a small fraction of basis states. At the same time, even simple NNs seem to have no problem in generalizing amplitudes from the training dataset onto the entire Hilbert space. They also have  very good capacity to express both sign and amplitude distributions of the studied states. Hence, in a search for neural quantum state architectures that can be trained to approximate the ground state of a large-scale many-body Hamiltonian, one should mainly focus on NNs that are at least capable of generalizing the ground state sign structure of moderately sized test systems. At this point, it is hard to give a concrete recipe of how to look for such architectures, but one of the possible ways could be to incorporate symmetries of the system into the network structure (improving the learning protocols may also help[50]). In our examples, using convolutional NNs that respected translational symmetries of the square and triangular lattices helped improve generalization quality significantly. This also suggests that our results are heuristic: although we have studied several most popular NN architectures, we cannot exclude a possibility that for certain other designs, the generalization will show features, qualitatively different from our findings.

Another important feature we have discovered is the threshold behavior of generalization as a function of the training dataset size. This is rather unusual and different from the smooth behavior known for standard models of learning, such as teacher–student scenario in a binary perceptron[47,48] and some other studies of NNs generalization[51]. From the point of view of vMC applications, it is desirable to understand how the required number of samples depends on system parameters, such as size and degree of frustration and training algorithm parameters.

As a closely related phenomenon, let us mention the fact known from the binary perceptron problem: bias towards dense clusters of local minima on the loss landscape makes generalization error a significantly steeper function of the number of the samples[49]. This may partially explain the observed abrupt change in generalization quality since stochastic gradient descent employed by us is known to have similar properties (bias towards wide minima of the loss landscape[40,52,53]). It would be very useful to have analytically tractable models which show the threshold behavior of generalization.

Finally, it is worth mentioning that, while the dip in generalization is not desirable in the context of variational energy optimization, it could be used as a tool to identify—in a completely unsupervised manner—the position of the phase transitions, similarly in spirit to approaches of refs. [54–57].

## Methods

**Training procedure**. In this study, we use feed-forward networks of three different architectures (dense 1-layer, dense 2-layer, and convolutional two-layer) to encode wave function coefficients via splitting them into amplitudes and signs. All of our networks have the same input format: spin configuration $|\mathcal{S}_i\rangle = |\sigma_1 \sigma_2 \dots \sigma_N\rangle$ represented as a binary sequence, $\sigma_k = \pm 1$. Network encoding amplitudes outputs a real number—natural logarithm of the amplitude. Network encoding sign structure outputs a probability $p \in [0, 1]$ for the corresponding sign to be plus. For inference we then use "+" whenever $p \geq 0.5$ and "−" otherwise. Thus, unlike the approach of ref. [16], we represent wave function signs using a binary classifier.

Both networks are trained on data obtained from exact diagonalization. We sample $\varepsilon_{\text{train}} \cdot K$ spin configuration from the Hilbert space basis according to

probability distribution $P(i) = \frac{|\psi_i|^2}{\sum_j |\psi_j|^2}$. They constitute the training dataset. Then, we sample another $\varepsilon_{\text{val}} \cdot K$ spin configurations which we use as a validation dataset during training. It enables us to monitor the progress and employ regularization techniques such as early stopping. In practical applications of NQS[16,26,29], SR[17,58], stochastic gradient descent[12,59], or generalized Lanczos[36], the training dataset is generated by Monte Carlo sampling from basis of the Hilbert space of the model, and, since dimension of the latter grows exponentially with the number of spins, only a tiny fraction of it can be covered with a Monte Carlo chain in reasonable time. Therefore, it is natural to mimic this incomplete coverage with $\varepsilon_{\text{train}}, \varepsilon_{\text{val}} \ll 1$.

To assess the performance of the NNs we evaluate overlap (scalar product) between exact eigenstate and the trial state. A trial state for sign NN is defined as a state with amplitudes from ED and sign structure encoded in a NN. Analogously, a trial state for amplitude NN is obtained by superimposing the exact sign structure onto the positive amplitudes encoded in the amplitude NN.

We train the classifier by minimizing binary cross-entropy loss function

$$\mathcal{L}^{\text{S}} = -\sum_i \left( \frac{1 + s_i}{2} \log p_i + \frac{1 - s_i}{2} \log (1 - p_i) \right), \qquad (3)$$

where $p_i$ is the predicted probability for the spin configuration $|\mathcal{S}_i\rangle$ to have sign $+1$, $s_i = \pm 1$ is the expected sign obtained from ED, and the sum is taken over the training dataset.

Training of the neural network which approximates amplitudes occurs via minimization of

$$\mathcal{L}^{\text{A}} = \sum_i \left( \log |\psi_i| - \log |\psi_i^e| \right)^2, \qquad (4)$$

where $\psi_i^e$ is the exact value of $i$th coefficient.

Usually, in ML algorithms it is crucial to choose hyperparameters correctly. For example, dependence of critical $\varepsilon_{\text{train}}$ on batch size is non-monotonic. Choosing a wrong batch size can lead to an order of magnitude increase of required $\varepsilon_{\text{train}}$. In our calculations, we typically work with batches of 64 or 128 samples. For optimizaion, we mostly use Adam[60] (a stochastic gradient-based method) with learning rates around $10^{-4}$–$10^{-3}$. Early stopping is our main regularization technique, but we have also experimented with dropout layers (which randomly throw away some hidden units) and $L_2$-regularization.

## Data availability

All simulations were carried out using PyTorch neural network manipulation package[61]. Original simulation results are available from the corresponding authors on a reasonable request.

## Code availability

Code to carry out the analysis is publicly available at http://github.com/nikita-astronaut/nqs_frustrated_phase and http://github.com/twesterhout/nqs-playground.

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

## Acknowledgements

We are thankful to Dmitry Ageev and Vladimir Mazurenko for collaboration during the early stages of the project. We have significantly benefited from encouraging discussions with Giuseppe Carleo, Juan Carrasquilla, Askar Iliasov, Titus Neupert, and Slava Rychkov. The research was supported by the ERC Advanced Grant 338957 FEMTO/NANO and by the NWO via the Spinoza Prize. The work of A.A.B. which consisted of designing the project (together with K.S.T.), implementation of prototype version of the code, and providing general guidance, was supported by Russian Science Foundation, Grant no. 18-12-00185. The work of N.A. which consisted of numerical experiments, was supported by the Russian Science Foundation Grant no. 16-12-10059. N.A. acknowledges the use of computing resources of the federal collective usage center Complex for Simulation and Data Processing for Mega-science Facilities at NRC "Kurchatov Institute", http://ckp.nrcki.ru/. K.S.T. is supported by Alexander von Humboldt Foundation and by the program 0033-2019-0002 by the Ministry of Science and Higher Education of Russia.

## Author contributions

A.A.B. and K.S.T. conceived the basic idea and designed the project. A.A.B. has created the prototype version of the code. K.S.T. contributed with his expertise in exact diagonalization. T.W. is responsible for the implementation of the main code, and N.A. has performed extensive numerical simulations. M.I.K. provided general guidance. All authors participated in discussions and contributed to the writing of the paper.

## Competing interests

The authors declare no competing interests.
