## [Peer Review File · Nature Communications]

Reviewers' comments:

Reviewer #1 (Remarks to the Author):

The authors use supervised learning to study the difficulties of applying Neural Quantum States (NQS) to frustrated magnets and propose that generalization in the prediction of the sign structure is the factor that limits the application of NQS ideas to challenging problems in frustrated magnetism. The ideas presented in the paper are common in machine learning but their use in condensed matter and frustrated magnetism is novel. I find the paper insightful and in my opinion deserves publication but I would like the authors to reasonably address the following issues:

- The training strategy should be discussed in the main text, even if briefly. Otherwise the statement of results is very unclear since understanding the setting is required to understand the results. I understand that these details are discussed in the methods, but having to read the methods to understand the results is inconvenient. For instance, explaining how the authors select the 1% fraction of Hilbert space for training, which can take one line, would make understanding the proposed results much easier.
- What do the authors mean by "as tensor networks [13], do not require stochastic Monte Carlo sampling and are thus amenable to exact optimization." What do authors mean by exact optimization? Optimization in tensor networks is not exact, there are approximations used in those techniques as well.
- Electric conductivity is typically very hard to get from variational calculations, so what do authors mean by this? Can they add a reference?
- One important criticism is that the results from the training setting used in this work may not necessarily translate to the most interesting setting which is energy optimization through VMC. Can the authors comment on the extension of these results to VMC setting? It seems reasonable to me that the results would still hold but it is not obvious to me that the suggestions in the conclusions are warranted. Doing VMC experiments is very easy and same predictions proposed in this work obtained, so I would suggest to run even small VMC calculations to confirm this scenario in the VMC setting.
- Why not CNN for triangular and kagome lattices since their results look so promising for the square lattice?
- Why don't the authors include symmetries (other than translations per their CNN experiments) which are known to help with this type of calculations?
- What do authors mean by "We believe that experiments of this kind would help to choose proper architectures to be used in iterative diagonalization schemes." Can they provide a reference to this?
- How do authors distinguish between expressibility and generalization? The authors state that expressibility is not an issue but show very little evidence that this is true.
- The origin of the improved performance of the CNN is not discussed. Can the authors discuss in the text why CNN is much better? I guess without details of the CNN architecture it is impossible to know if their ansatz has translational symmetry but if it does, it could be the reason.
- Does figure 3 saturate or keeps improving if you keep going for larger epsilon?
- A more interesting question is how does figure 2,3 scale with system size. Can the authors comment on the size scaling of the overlap after training? Does it get worse or does it stay the same as the size of the system increases?
- Please explain the meaning of shades in plots.
- The conclusion of the paper sounds plausible but the authors have not shown enough evidence that their conclusions apply to energy minimization, which is where these results would be the most relevant. Note that in that setting, the "cost function" as well as the training dynamics are completely different from the experiments in this paper. Both sign and amplitude are concomitantly predicted in VMC without any information or data other than the Hamiltonian. It does not sound crazy to suggest these conclusions in the light of their experiments but it seems to

me that the conclusion are stronger than the evidence presented in this paper suggests. For instance, the test the authors proposed "A neural quantum state can be trained to approximate ground state of a large-scale many-body system only if it is capable of generalizing sign structure of a moderately- sized (exactly solvable) system ground state." should be substantiated/supported by variational Monte Carlo simulations.

- A binary classifier predicts the probability P of an input S of belonging to a class a or b . Related to this, how do authors compute the sign in the figures? By using samples from P or by assigning the most likely outcome? Does the computed P affects the amplitudes of the wave function?
- Other architectures and training strategies may invalidate these results. For instance, a better choice of NN with all the symmetries of the Hamiltonian could be used, which very likely would improve over these results. May the authors please add a statement explaining that these results are heuristic and not a proof, and thus it could be that by including a better choice of NN these results may be overcome.

Minor:

- Stick to one type of English in the text (eg generalization vs generalisation)
- Define ϵ_{train} when first mentioned.
- "desired matrix elements can be evaluated in a time polynomial in the number of particles" This is not true, what's true is that the evaluation of the heuristics people use are efficient but the MC methods are not efficient or have theoretical guarantees.

Reviewer #2 (Remarks to the Author):

The paper focuses on the generalization property of Neural Quantum States. The Neural Network States are new kinds of variational wave function ansatz, inspired by deep learning neural networks. Most previous papers are focusing on the expressive power of the neural network regarding topological properties or area law structure, from Restricted Boltzmann Machine to Convolutional Neural Network. People are mostly comparing ground state energy.

However, not too much attention was paid to the generalization property, which measures the difference between generated results with the real wave function. Generalization is a very important point to consider in machine learning. The authors have made four important claims.

1. Generalization from a relatively small subset of Hilbert space basis of the wave function sign structure is not granted even when the ansatz is able to express the ground state with high accuracy. Very well known to machine learning practitioners, this fact is also valid for spin systems, in both frustrated and ordered phases.
2. Construction and training of a network to achieve good generalization, a task which is relatively simple in the ordered phase, becomes much harder upon approaching the frustrated phase.
3. Quality of generalization depends on the size of the training set in an abrupt way exhibiting a sharp increase at some ϵ_{train} .
4. Prediction of wave function amplitudes turns out to be a substantially easier task than prediction of signs.

The authors have calculated the J1-J2 models on three different lattices, square, triangular and kagome. The exact wave function comes from exact diagonalization. By learning the signs and amplitudes separately, the authors clearly prove claim 4 (Fig.4). The generalization is characterized by the overlap between the learned wave function out from different training set and exact wave function elements. The numerical results clearly show there is a minimum size of the

training set to get a reasonable result (Fig.3 Fig.S2 Fig S4 and Claim 3). By tuning J_2 , the strength of frustration, figure 2 has shown that the sign problem is severe in the frustrated region.

All the claims have clear and corresponding numerical proof.

However, the system size is too small to make a general conclusion, especially for the calculation of ϵ_{train} value. I think a larger system size by ED or other approximate numerical methods, which gives wave function, are recommended. It may introduce some errors for the large amplitudes and sign errors for small amplitudes. So only comparing the wave function elements with large amplitudes is a choice.

The codes are given online, which is easy for other researchers to reproduce.

The paper is well organized and the logic is clear.

In conclusion, the paper is studying a very important, however, somewhat neglected aspect of Neural Network State. However, some of the claims need more numerical support to make it better. I would suggest the publication when larger system size data are given.

Reviewer #3 (Remarks to the Author):

In this work, the authors studied the ability of neural networks to generate and predict wavefunctions of frustrated systems, by considering a square, a triangular, and a kagome magnet. They discussed the importance of "generalization" and its behaviors with increasing frustration. Their discussions are systematic; nevertheless, the results are expected. It is well known that frustration can lead to many nearly degenerate (low-lying) states. To correctly represent the wavefunction of a frustrated system, high precision and good convergence are demanded, in both traditional methods, such as variational Monte Carlo and DMRG, and machine learning techniques. It is not surprising that neural networks have poor performance there. It does not seem to me that the manuscript contains sufficient new insights.

A few suggestions are listed below, which may be considered for improving the presentation.

- 1) The meaning of "generalization" and its relation/distinction to "expressibility" and "prediction" should be clearly defined in Introduction.
- 2) ϵ_{train} is not introduced before using.
- 3) There is ambiguity about the term "frustrated phases". The authors seem to mean (magnetically) disordered phases. However, a highly frustrated Hamiltonian can also develop a local order. Frustration can be used to describe interactions or parameter regions but not to specify the nature of a phase.
- 4) The review of machine learning methods' ability and achievements in the Introduction sounds misleading. Most of them still heavily rely on fine-tuning and are only applicable to simple problems.

Dear Editor,

We are happy that Reviewers #1 and #2 find that our paper is "insightful", "well organized" and that it "deserves publication" with reservation by the Reviewer #2 that additional data have to be given for larger system sizes. We have reworked the paper, most importantly, a) considering much larger system sizes (up to 36 spins instead of 24 spins in the previous version) and b) performing additional numerical experiments to better illustrate the relation of our findings to the very important method of fitting trial wavefunctions, namely Stochastic Reconfiguration. We have also improved the presentation, following numerous suggestions by all reviewers. Below, we provide point-by-point reply to Reviewers' comments. We believe that our paper benefited a lot from Reviewers suggestions and hope that it can now be published.

Kind regards,

T. Westerhout, N. Astrakhantsev, K. Tikhonov, M. I. Katsnelson, and A. A. Bagrov

Reviewer #1

Reviewer's comment:

The training strategy should be discussed in the main text, even if briefly. Otherwise the statement of results is very unclear since understanding the setting is required to understand the results. I understand that these details are discussed in the methods, but having to read the methods to understand the results is inconvenient. For instance, explaining how the authors select the 1% fraction of Hilbert space for training, which can take one line, would make understanding the proposed results much easier.

Our reply:

We have added more details on the training procedure and on how the training set is selected in the main text, see page 3 after Eq. (2): "First, we solve each of the models using exact diagonalization. Then, with the exact ground state as target, we use supervised learning to train the NQS. During the training procedure, NNs are shown only a tiny fraction of the ground state (which is chosen by sampling from the probability distribution $\propto |\psi_i|^2$). Quality of the approximation is then assessed on the remaining part of the Hilbert space basis which we call test dataset. We tune both the training dataset size as well as the degree of frustration (controlled by J_2/J_1)."

Reviewer's comment:

What do the authors mean by "as tensor networks [13], do not require stochastic Monte Carlo sampling and are thus amenable to exact optimization." What do authors mean by exact optimization? Optimization in tensor networks is not exact, there are approximations used in those techniques as well.

Our reply:

We have edited the respective comment to make it more precise: “Certain tensor network variational ansätze, e.g. Matrix Product States [13], do not require stochastic Monte Carlo sampling and are thus amenable to exact optimization.”

Reviewer's comment:

Electric conductivity is typically very hard to get from variational calculations, so what do authors mean by this? Can they add a reference?

Our reply:

The Reviewer is right. We have dropped the remark on the electric conductivity.

Reviewer's comment:

One important criticism is that the results from the training setting used in this work may not necessarily translate to the most interesting setting which is energy optimization through VMC. Can the authors comment on the extension of these results to VMC setting? It seems reasonable to me that the results would still hold but it is not obvious to me that the suggestions in the conclusions are warranted. Doing VMC experiments is very easy and same predictions proposed in this work obtained, so I would suggest to run even small VMC calculations to confirm this scenario in the VMC setting.

Our reply:

This is a very important question. In order to confirm that our conclusions hold in the VMC setting, we have performed SR calculations for all three models (triangle, square, Kagome). On Fig. 2, lower panel, we show that overlap of optimized wavefunctions with the ground state behaves as a function of J_2/J_1 in a way which is qualitatively very similar to that of the generalization quality. We have also added additional note on Stochastic Reconfiguration, see end of page 4: “We believe that experiments of this kind would help to choose proper architectures to be used in vMC methods such as SR. In Stochastic Reconfiguration scheme, parameter updates are calculated using a small (compared to the Hilbert space dimension) set of vectors sampled from the probability distribution proportional to $|\psi|^2$. This closely resembles the way we choose our training dataset. Moreover, SR does not optimize energy directly, rather at each iteration it tries to maximize the overlap between the NQS $|\Psi\rangle$ and the result of its imaginary time evolution $(1-\delta t \hat{H})|\Psi\rangle$. Hence, even though our supervised learning scheme and SR differ drastically, their efficiencies are strongly related. To make this correlation more apparent, we have performed several vMC experiments for 24-spin clusters. In two panels of Fig.2, we compare results of running the two learning schemes for different values of J_2/J_1 , and show that they follow similar patterns”

Reviewer's comment:

Why not CNN for triangular and kagome lattices since their results look so promising for the square lattice?

Our reply:

For Kagome, it is non-trivial to implement the CNN architecture since the structure of the network filters does not conform to the geometry of the lattice. Thus we refrained from studying this case. We have added results for CNN for triangular lattice, see Fig. 2 upper middle panel.

Reviewer's comment:

Why don't the authors include symmetries (other than translations per their CNN experiments) which are known to help with this type of calculations?

Our reply:

Indeed, we believe that account for other symmetries would be beneficial for generalization properties of the NN ansatz. It is probable, for example, that upon account for the reflection symmetries, the critical value of ϵ would shift to the lower values. We do not expect, however, that this will change any of our qualitative conclusions (such as threshold character of generalization or its correlation to SR performance).

Reviewer's comment:

What do authors mean by "We believe that experiments of this kind would help to choose proper architectures to be used in iterative diagonalization schemes." Can they provide a reference to this?

Our reply:

We suspect that Reviewer's question might have been caused by our statement which was not sufficiently accurate. We rephrased this piece of text and added additional note on Stochastic Reconfiguration, see end of page 4.

Reviewer's comment:

How do authors distinguish between expressibility and generalization? The authors state that expressibility is not a issue but show very little evidence that this is true.

Our reply:

We added a bit more formal definitions of expressibility and generalization to Introduction. Numerical support for our statements that expressibility is relatively simple and is shown on the Fig. 2, upper panel. See also the comment in the main text, page 5: "For clusters of 24 spins we have trained the networks on the entire ground state and

found that expressibility of the ansatz is not an issue, -- we could achieve overlaps above 0.94 for all values of J_2/J_1 ”

Reviewer's comment:

The origin of the improved performance of the CNN is not discussed. Can the authors discuss in the text why CNN is much better? I guess without details of the CNN architecture it is impossible to know if their ansatz has translational symmetry but if it does, it could be the reason.

Our reply:

It is very plausible that superior performance of CNN is related to imposing translational symmetry, see the added comment at the end of page 4: “Such good performance is most likely due to the fact that our implementation of CNNs accounts for translational symmetry (see Supplementary Material for an in-depth explanation of used NN architectures)”

Reviewer's comment:

Does figure 3 saturate or keeps improving if you keep going for larger epsilon?

Our reply:

In the previous version where relatively little systems have been studied, we had to stop the plot of eps-dependence. Continuing to the region of larger eps would require calculation of overlap on the part of the wavevector containing too small weight (due to exhaustion of the set of the large-weight wavefunction amplitudes), see footnote 3: “It is hard to see this plateau in the system of 24 spins as it requires too large "train, such that all relevant basis vectors end up in the training dataset, and overlap computed on the rest of the basis is meaningless.”. For larger system sizes we are able to show that the overlap indeed shows a plateau as function of eps, see Fig. S7.

Reviewer's comment:

A more interesting question is how does figure 2,3 scale with system size. Can the authors comment on the size scaling of the overlap after training? Does it get worse or does it stay the same as the size of the system increases?

Our reply:

This is a very interesting question. We have added Fig. 4 with system-size dependence of the “critical” size of the training set.

Reviewer's comment:

Please explain the meaning of shades in plots.

Our reply:

The shades in plot mean character of the phases, as indicated in the captions. In the previous version of the paper, we have also used shades to indicate error margins, which are now shown by more conventional error bars.

Reviewer's comment:

The conclusion of the paper sounds plausible but the authors have not shown enough evidence that their conclusions apply to energy minimization, which is where these results would be the most relevant. Note that in that setting, the “cost function” as well as the training dynamics are completely different from the experiments in this paper. Both sign and amplitude are concomitantly predicted in VMC without any information or data other than the Hamiltonian. It does not sound crazy to suggest these conclusions in the light of their experiments but it seems to me that the conclusion are stronger than the evidence presented in this paper suggests. For instance, the test the authors proposed “A neural quantum state can be trained to approximate ground state of a large-scale many-body system only if it is capable of generalizing sign structure of a moderately-sized (exactly solvable) system ground state.” should be substantiated/supported by variational Monte Carlo simulations.

Our reply:

In the revised paper, we have added variational Monte Carlo experiments (Stochastic Reconfiguration), see Fig. 2, lower panel for the results. We have also added additional note on Stochastic Reconfiguration, see end of page 4.

Reviewer's comment:

A binary classifier predicts the probability P of an input S of belonging to a class a or b . Related to this, how do authors compute the sign in the figures? By using samples from P or by assigning the most likely outcome? Does the computed P affects the amplitudes of the wave function?

Our reply:

We set the sign to the most probably outcome and the computed P does not affect the amplitude. We added the corresponding comment to the text: “(the sign is chosen by following the most probable outcome according to the NN)”

Reviewer's comment:

Other architectures and training strategies may invalidate these results. For instance, a better choice of NN with all the symmetries of the Hamiltonian could be used, which very likely would improve over these results. May the authors please add a statement explaining that these results are heuristic and not a proof, and thus it could be that by including a better choice of NN these results may be overcome.

Our reply:

We have added a clarifying statement in the end of page 7: “This also suggests that our results are heuristic: although we have studied several most popular NN architectures, we can not exclude a possibility that for certain other designs, the generalization will show features, qualitatively different from our findings.”

Reviewer's comment:

Stick to one type of English in the text (eg generalization vs generalisation)

Our reply:

We worked on English to make it coherently of US type throughout the paper.

Reviewer's comment:

Define `epsilon_train` when first mentioned.

Our reply:

In the new version of the text, we define `epsilon_train` on the first appearance: item (iii) in the beginning of the Results section.

Reviewer's comment:

“desired matrix elements can be evaluated in a time polynomial in the number of particles” This is not true, what’s true is that the evaluation of the heuristics people use are efficient but the MC methods are not efficient or have theoretical guarantees.

Our reply:

We agree that the paragraph to which the Reviewer refers, was not sufficiently clearly written. We have dropped it and instead added the corresponding comments to the discussion of the Stochastic Reconfiguration method (which we added to clarify the relation between our findings and performance of vMC).

Reviewer #2

Reviewer's comment:

However, the system size is too small to make a general conclusion, especially for the calculation of `strain` value. I think a larger system size by ED or other approximate numerical methods, which gives wave function, are recommended. ... In conclusion, the paper is studying a very important, however, somewhat neglected aspect of Neural Network State. However, some of the claims need more numerical support to make it better. I would suggest the publication when larger system size data are given.

Our reply:

In the new version of the manuscript, we added more data for larger system sizes, which

now extend from 24 to 36 spins with all our conclusions remaining qualitatively valid. See, for example, Fig. 4 with size dependence of the critical size of the train set and Figs. S5 and S7 for more details on systems of size 30 and 36.

Reviewer #3

Reviewer's comment:

To correctly represent the wavefunction of a frustrated system, high precision and good convergence are demanded, in both traditional methods, such as variational Monte Carlo and DMRG, and machine learning techniques. It is not surprising that neural networks have poor performance there.

Our reply:

It is true that high precision and good convergence are very important for representation of a quantum state of the frustrated system. In our study we demonstrate that in addition to these issues, one has to take care of generalization properties of the underlying ansatz (the latter being very different from the former). We do not agree, however, that neural networks have poor performance for representing the states of the frustrated systems: all these issues do not preclude a possibility for NN to perform on par with state-of-the-art methods in this regime (see 1903.06713 as an example). We believe that understanding of all the issues may help improve this approach further.

Reviewer's comment:

The meaning of "generalization" and its relation/distinction to "expressibility" and "prediction" should be clearly defined in Introduction.

Our reply:

We extended the introductory discussion of the concepts "expressibility" and "generalization" on the 2nd page of the manuscript. We substantially reduced the usage of the term "prediction" to avoid confusion.

Reviewer's comment:

ϵ_{train} is not introduced before using.

Our reply:

We fixed it to define it on the first appearance: item (iii) in the beginning of the Results section.

Reviewer's comment:

There is ambiguity about the term "frustrated phases". The authors seem to mean (magnetically) disordered phases. However, a highly frustrated Hamiltonian can also

develop a local order. Frustration can be used to describe interactions or parameter regions but not to specify the nature of a phase.

Our reply:

We agree with the referee and made corresponding changes to the wording in the text and added the following clarifying comment: “These models are known to host spin liquid phases in certain domains of J_2/J_1 , to which we further refer as frustrated regions.”

Reviewer's comment:

The review of machine learning methods' ability and achievements in the Introduction sounds misleading. Most of them still heavily rely on fine-tuning and are only applicable to simple problems.

Our reply:

We have adjusted the introduction to make it a bit more modest.

REVIEWERS' COMMENTS:

Reviewer #1 (Remarks to the Author):

The authors have addressed all my comments and request for experiments satisfactorily. I recommend the paper for publication in nature communications.

Juan Felipe Carrasquilla Álvarez

Reviewer #2 (Remarks to the Author):

The authors had extended the system size from 24 to 36, which makes it more convincing. I suggest the publication of the paper.